# Modified Lactoperoxidase System as a Promising Anticaries Agent: In Vitro Studies on *Streptococcus mutans* Biofilms

**DOI:** 10.3390/ijms241512136

**Published:** 2023-07-28

**Authors:** Marcin Magacz, Sergio Alatorre-Santamaría, Karolina Kędziora, Kacper Klasa, Paweł Mamica, Wiktoria Pepasińska, Magdalena Lebiecka, Dorota Kościelniak, Elżbieta Pamuła, Wirginia Krzyściak

**Affiliations:** 1Department of Medical Diagnostics, Faculty of Pharmacy, Jagiellonian University Medical College, Medyczna 9, 30-688 Kraków, Poland; marcin.magacz@uj.edu.pl (M.M.); karolina.kedziora@gmail.com (K.K.); kacper.klasa@student.uj.edu.pl (K.K.); paw.mamica@student.uj.edu.pl (P.M.); wiktoria.pepasinska@student.uj.edu.pl (W.P.); magdalena.lebiecka@student.uj.edu.pl (M.L.); 2Doctoral School of Health and Medical Sciences, Jagiellonian University Medical College, św. Łazarza 16, 31-008 Kraków, Poland; 3Department of Biotechnology, Biological Science Division, Autonomous Metropolitan University, San Rafael Atlixco 186, Mexico City 09310, Mexico; salatorre@izt.uam.mx; 4Department of Pediatric Dentistry, Institute of Dentistry, Jagiellonian University Medical College, Montelupich 4, 31-155 Krakow, Poland; dorota.koscielniak@uj.edu.pl; 5Department of Biomaterials and Composites, Faculty of Materials Science and Ceramics, AGH University of Science and Technology, Al. Mickiewicza 30, 30-059 Kraków, Poland; e.pamula@agh.edu.pl

**Keywords:** lactoperoxidase, *Streptococcus mutans*, biofilm, dental caries, lactoperoxidase system modulators, cariogenic biofilm

## Abstract

The lactoperoxidase (LPO) system shows promise in the prevention of dental caries, a common chronic disease. This system has antimicrobial properties and is part of the non-specific antimicrobial immune system. Understanding the efficacy of the LPO system in the fight against biofilms could provide information on alternative strategies for the prevention and treatment of caries. In this study, the enzymatic system was modified using four different (pseudo)halide substrates (thiocyanate, thiocyanate-iodide mixture, selenocyanate, and iodide). The study evaluated the metabolic effects of applying such modifications to *Streptococcus mutans*; in particular: (1) biofilm formation, (2) synthesis of insoluble polysaccharides, (3) lactate synthesis, (4) glucose and sucrose consumption, (5) intracellular NAD^+^ and NADH concentrations, and (6) transmembrane glucose transport efficiency (PTS activity). The results showed that the LPO–iodide system had the strongest inhibitory effect on biofilm growth and lactate synthesis (complete inhibition). This was associated with an increase in the NAD^+^/NADH ratio and an inhibition of glucose PTS activity. The LPO–selenocyanate system showed a moderate inhibitory effect on biofilm biomass growth and lactate synthesis. The other systems showed relatively small inhibition of lactate synthesis and glucose PTS but no effect on the growth of biofilm biomass. This study provides a basis for further research on the use of alternative substrates with the LPO system, particularly the LPO–iodide system, in the prevention and control of biofilm-related diseases.

## 1. Introduction

Numerous public health reports, both global and from different regions, state that dental caries is among most prevalent chronic diseases in the world. It is estimated that the disease affects around 2.4 billion people worldwide [1]. The progression of dental caries is considered to be the result of four major factors: (i) cariogenic microorganisms that form biofilms, such as *Streptococcus mutans*; (ii) dietary sugars (mainly glucose, fructose, and sucrose); (iii) host susceptibility; and (iv) time [2,3,4].

The key step in the development of dental caries is the formation of a biofilm, a complex structure consisting of bacterial cells and various types of polymeric substances. The biofilms formed on tooth surfaces in vivo are multi-species biofilms composed of commensal and cariogenic microorganisms. Under conditions favourable to the formation of cariogenic biofilms, dysbiosis occurs in the oral cavity, leading to the proliferation of cariogenic microorganisms [5]. The process of biofilm formation can be divided into several phases. The first phase is adhesion, where bacterial cells attach to the surface using the adhesive substances they produce. Initially, the cells do not adhere permanently to the surface, but under favourable conditions, they attach permanently and start the growth phase. At this stage, due to the undeveloped biofilm matrix, this process can be effectively stopped by the use of chemotherapeutic agents (e.g., chlorhexidine) or various plant extracts found in oral hygiene products [6]. During the growth phase, the number of cells increases, and the synthesis and secretion of extracellular polymeric substances begins. These substances form a matrix around the bacterial cells, providing structural support for the biofilm. The matrix is rich in polysaccharides, proteins, DNA, and other organic compounds that help maintain the structure of the biofilm. Over time, the biofilm grows and matures. Bacterial cells in a biofilm can use chemical signals to communicate with each other, which allows them to coordinate their activities. A mature biofilm can have a complex structure, with different layers of cells and a matrix. In the final stage of the biofilm life cycle, the cells within the structure can be released into the environment. This process is known as biofilm spreading and can lead to infection and colonization of new surfaces [7,8].

Cariogenic biofilm microorganisms are characterized by efficient carbohydrate metabolism, resulting in the production of large amounts of organic acids and components of the biofilm matrix, as well as high resistance to acidic environments. One microorganism that meets these criteria is *S. mutans*, considered one of the main etiological factors of dental caries and a model organism in in vitro studies on cariogenic biofilms [9]. In light of the mentioned cariogenic factors of this microorganism, bioactive molecules with anti-caries activity targeted against biofilms should demonstrate the ability to inhibit carbohydrate transport processes into the bacterial cells, glycolysis, and biofilm biomass growth. The efficiency of chemotherapeutic agents against microorganisms within the biofilm decreases as the biofilm matures and the matrix develops. Biofilm microorganisms are 10 to 1000 times more resistant to antimicrobial agents than planktonic microorganisms due to impeded penetration through the biofilm structure. The use of conventional antimicrobials against microorganisms within biofilms may require the use of these substances at concentrations that are unacceptable due to toxicity. Currently, the only effective way to eliminate cariogenic biofilm outside the dental office is mechanical removal during tooth brushing. Other important measures in caries prevention include fluoride prophylaxis (toothpaste, mouthwashes, varnishes), the use of preparations that promote tooth remineralisation, and, finally, systematic dental check-ups, including treatment of cavities with fillings if necessary [6,10]. This situation has led to a search for different ways of prevention and treatment. One area of interest covers research on antimicrobial biomolecules and particularly those having antibiofilm properties. This approach aims at limiting drug resistance development, minimising side effects, and achieving efficacy only against pathological components of the oral cavity microbiome without interfering with the commensal microbiome [11].

The biomolecule that meets caries prevention criteria is lactoperoxidase (LPO), a heme peroxidase that oxidises (pseudo)halogen ions to products with antimicrobial properties at the expense of hydrogen peroxide through the halogenation cycle. These products are characterised by high redox potential and the ability to impair the biological function of proteins and nucleotides of microorganisms by oxidation of these molecules. LPO, together with its substrates and products, forms the LPO system. This enzyme is excreted by epithelial cells into multiple biological fluids, including saliva, tears, airway surface fluid, female reproductive tract fluid, milk, and others, where the enzyme plays an important role in innate immunity [12,13]. In the oral cavity, this enzyme plays a role in regulating the composition of the microbiome and has a biocidal or biostatic effect on viruses and pathogenic species of bacteria and fungi [13]. It should be noted that the efficiency of the LPO system is related to individual factors, mainly salivary flow and genetic factors, which cause high variability in salivary LPO activity [14]. The characteristic composition of saliva means that the main substrate for LPO in vivo is the thiocyanate ion (SCN^−^), which is present in saliva in relatively high concentrations (0.5–2 mM) [15] (LPO, unlike myeloperoxidase, does not oxidise the chloride ion) and oxidised to hypothiocyanite (OSCN^−^)/hypothiocyanous acid (HOSCN) [16]. The research conducted over the years has found that LPO is characterised by a wider substrate spectrum, and if other (pseudo)halide ions are available, they can also enter the halogenation cycle. Although the antimicrobial action of LPO systems where the (pseudo)halide substrates are either SCN^−^ or I^−^ (the ions are present in the saliva only in trace quantities, insufficient to generate a product in a quantity that would be effective against microorganisms) is well described with regard to planktonic microorganisms, the effectiveness against microorganisms located in biofilm structures has not yet been investigated. Furthermore, detailed data on both the antimicrobial and antibiofilm activities of other possible LPO products are lacking; in particular, the newly described systems LPO–selenocyanate (SeCN^−^) (substrate not physiologically present in body fluids) with hyposelenocyanite (OSeCN^−^) as the product and LPO–thiocyanate–iodide, which is capable of cyanogen iodide (ICN) synthesis [13,17,18].

Toothpastes and mouthwashes containing bovine LPO and other biomolecules (lactoferrin, lisozyme, bovine milk immunoglobulin) are currently available on the market. In order to increase the access of the (pseudo)halide substrate to the LPO system, these products are enriched with exogenous SCN^−^ ions. The limiting factor for the production of reactive LPO system products in the oral cavity is the availability of H_2_O_2_, which is synthesised in vivo by both host oxidases and some oral streptococci [19]. For this reason, products are also enriched with enzymatic systems that generate H_2_O_2_ in situ, such as glucose oxidase or lactate oxidase. The studies currently available on this type of product, both in vitro studies and clinical trials, are inconclusive as to their effectiveness [13]. It seems important to carry out further research into the possibility of increasing the efficiency of the LPO system by using alternative substrates that can be oxidised to products with stronger antibiofilm properties than the physiological product. The potentially stronger antibiofilm properties of these systems in vivo may not only be a result of the stronger antibiofilm properties of the product. On its own, an externally supplied (pseudo)halide substrate that is not present in saliva can, due to its low molecular weight, potentially penetrate into biofilms and become a substrate for endogenous LPO adsorbed within the biofilm structure during its growth, leading to the production of reactive LPO products within the biofilm.

The aim of the study was to provide preliminary in vitro verification of the antibiofilm potential of four LPO system modifications, each with a different (pseudo)halide substrate. The focus was on the assessment of key *S. mutans* biofilm characteristics that are directly responsible for dental caries development; namely: biomass growth capacity, extracellular matrix synthesis, lactic acid synthesis from glucose, and sucrose consumption. In addition, intracellular levels of NAD^+^ and NADH and phosphotransferase system (PTS) activity were evaluated to compare the mechanism of action of all systems in relation to carbohydrate metabolism.

## 2. Results

### 2.1. LPO System Influence on Total Biomass and Insoluble Polysaccharide Mass

In the case of biomass, during the insoluble polysaccharide mass assay and lactate/glucose/sucrose kinetic assay, LPO system variants were used in the presence of three different H_2_O_2_ concentrations (Figure 1). This approach was employed to evaluate the relationship between H_2_O_2_ concentration and the observed effect and to select the optimal H_2_O_2_ concentration that would result in the highest LPO system antibiofilm activity but would not damage host cells.

The total biofilm biomass and the total biomass of the insoluble biofilm polysaccharide after exposure to each of the tested LPO system modifications are shown in Figure 1 and Appendix A. In this experiment, two-hour *S. mutans* biofilms (early biofilm formation phase) were exposed to the tested systems. The use of early-formation-phase biofilms made it possible to observe any further biofilm biomass increase.

The study revealed that the rate of reduction in biofilm biomass/insoluble exopolysaccharide mass was dependent on the type of substrate/product of the LPO system, the concentration of hydrogen peroxide, and, consequently, the amount of the product of the LPO system. The strongest statistically significant inhibitory effect on biomass growth was observed in the case of the LPO system with iodide as the substrate at the highest H_2_O_2_ (last red column in the first segment of Figure 1B). The second most effective system that significantly decreased biomass and insoluble polysaccharide production contained SeCN^−^ as a substrate. In the case of the physiological LPO system (SCN^−^ as substrate), no statistically significant reduction in the mass of the insoluble polysaccharide was observed. There were only slight statistically significant reductions in total biomass with two of the three tested concentrations of H_2_O_2_ (250 μM and 500 μM), respectively. The LPO system with a substrate mixture of SCN^−^ and I^−^ did not exhibit the ability to significantly reduce insoluble polysaccharide biofilm mass; however, 27% of the total biomass reduction was observed in the sample containing 500 μM H_2_O_2_.

Hydrogen peroxide alone has no statistically significant effect on insoluble polysaccharide mass, and only slight statistically significant reductions in total biomass (ca. 12%) were observed at 500 μM and 250 μM H_2_O_2_ concentrations. (Pseudo)halides alone did not influence total biomass and insoluble polysaccharide mass in biofilms tested (Tukey’s test *p* > 0.05) (Appendix A).

### 2.2. Lactate, Glucose, and Sucrose Metabolism in Biofilm

A simple and quick way to measure the effects of compounds with anticariogenic potential is to investigate parameters related to biofilm carbohydrate metabolism; more specifically, lactic acid production and biomass growth. In this study, lactate production and glucose and sucrose consumption were measured with the amperometric method utilising a biosensor (Figure 2). To the best of our knowledge, this is the first study on microorganisms that involved this measurement technology. This approach provides a fast, semi-automated, accurate measurement suitable for high-capacity analysis for screening.

The kinetic changes in the concentrations of the three compounds were measured. Figure 2A and Figure 3A show the changes in time in the lactate, glucose, and sucrose concentrations in the biofilm growth medium after treating the 2 h and 24 h biofilms with LPO systems for 15 min.

Of all the combinations tested, the strongest inhibition of lactate production was observed in the case of the LPO–I^−^ system in both the 2 h and 24 h biofilms (Figure 2B and Figure 3B). There, lactate synthesis was completely inhibited for the duration of the experiment. Promising results were also observed in the sample containing the LPO–SeCN^−^ system, where the relative AUC (lactate production) decreased significantly for all H_2_O_2_ concentrations tested, but the decrease reached higher values for the 24 h biofilm. A much smaller but statistically significant reduction in lactate production was only observed in the case of 24 h biofilms treated with the physiological LPO–SCN^−^ system, with relative decreases in the AUC ranging from 32% to 23% for the tested concentrations of H_2_O_2_. In samples with the LPO–SCN^−^+I^−^ system, the lactate production rate expressed as the relative AUC was statistically significant only for 1000 and 500 μM of H_2_O_2_ in 24 h biofilms.

Statistically significant decreases in sucrose consumption by biofilms treated with LPO systems were observed only in the case of the LPO-I^−^ system for the 2 h and 24 h biofilms (Figure 2C and Figure 3C, Appendix A). Glucose concentration remained constant both in the tested samples and in the control for the 2 h and 24 h biofilms (Figure 2A,C and Figure 3A,C).

### 2.3. Influence of LPO Systems on PTS Activity

The phosphoenolopuryvate PEP–carbohydrate phosphotransferase system is the most important transport system responsible for transmembrane mono- and disaccharide transport in *S. mutans*.

The presented studies were the first to investigate the effect of the LPO system on the PTS system and demonstrated that all tested LPO systems could inhibit the glucose PTS system (Figure 4 and Appendix A). The strongest inhibitory effect was observed in the case of the LPO–I^−^ system (85% inhibition compared to the control). The lowest (but still relatively high) inhibition was observed with the physiological LPO–SCN^−^ system (72% inhibition). In the case of the LPO enzyme and (pseudo)halide substrates alone, there were no statistically significant changes in glucose PTS activity.

### 2.4. Influence of LPO Systems on Biofilm NAD^+^/NADH

Analysis of NAD^+^ and NADH concentrations in biofilms after treatment with LPO systems showed that the most significant changes were observed in the case of the LPO–I^−^ system (Figure 5). In that case, a statistically significant increase in NAD^+^ concentration (6.40 ± 0.07 μg/well vs. 3.36 ± 0.56 μg/well in the control), a decrease in NADH concentration (0.78 ± 0.01 μg/well vs. 2.29 ± 0.19 μg/well in the control), and an increase in the NAD^+^/NADH ratio (8.220 ± 0.012 vs. 1.470 ± 0.231) were observed. There were no other statistically significant changes in NAD^+^ levels with other LPO systems. The NADH concentration increased slightly but statistically significantly after treatment with the LPO–SCN^−^+I^−^ system and LPO–SeCN^−^. The (pseudo)halide substrates and hydrogen peroxide alone did not cause statistically significant changes in any of the three parameters after incubation with biofilms (Appendix A).

## 3. Discussion

Over the last several years, a great deal of research has been published on the antimicrobial activity of LPO. Most of it has focused on the interactions of the physiological LPO substrate thiocyanate [20,21] and the non-physiological substrate iodide [22,23,24]. However, in recent years, there have been several studies reporting on other possible substrates and products of the LPO system [18,25]. In our study, in addition to the aforementioned systems, we investigated two other systems: the newly described LPO–selenocyanate system with hyposelenocyanite (OSeCN^−^) as a product [18] and a modified physiological system, the LPO-SCN^−^+I^−^ system, which is characterised by the production of highly toxic cyanogen iodide (ICN) [17]. Unfortunately, almost all of these previous studies focused on the antimicrobial activity of the LPO systems against planktonic cells and not on microorganisms in biofilms. Due to the barrier provided by the extracellular matrix of the biofilm, the greater buffering capacity for antimicrobial compounds, and the biochemical and genetic cooperation between the microorganisms that are part of the consortium attached in biofilms, this structure makes its inhabitants 10 to 1000 times more resistant to antimicrobials than planktonic cells [26]. In addition, many diseases, including dental caries, do not depend on the presence of planktonic cells but on the formation of biofilms, and successful eradication of biofilms is critical to the control of these diseases.

It should be emphasised that caries development in vivo is tied to the accumulation of a biofilm and the metabolic activity of numerous species of microorganisms, but *S. mutans* is considered to be one of the main etiological factors of this disease. This is linked to its strong adhesion to hard tooth tissue and its ability to create biofilms, produce organic acids, and thrive in an acidic environment [27]. For these reasons, the presented research was focused on conducting experiments on a single-species biofilm. This approach allowed us to eliminate the additional factors related to interspecies interaction found in multispecies biofilms that can intervene with the results. Moreover, using single-species biofilms means that the experiments can be replicated more easily to verify or extend the research [9,28].

In the presented research, bovine milk LPO was used due to its high level of similarity (structural and functional) to the human LPO molecule [13]. A buffered solution of 50 nmol/L LPO was used throughout the study that reflected the physiological enzyme concentration found in saliva [29]. All LPO reactions were performed in phosphate buffered solution (PBS) at pH 7.4, which reflected the environmental conditions of the oral cavity of a healthy person between meals. It should be noted that, in the case of the LPO–I^−^ and LPO–SCN^−^+I^−^ mixture, the resulting products were therefore pH-dependent. The pH of the oral cavity is not constant and can vary depending on the time of day, the presence of diseases, the time since the last meal and its composition, and, finally, the metabolic activity of the microorganisms. In the case of the product of the LPO system with SCN^−^+I^−^ as the substrate, the formation of highly toxic cyanogen iodide decreases with a decrease in environmental pH [25].

In the available studies, two approaches to exposing microorganisms to the LPO system can be found [29,30]. The first consists of creating a mixture of the substrates and the enzyme and triggering the reaction; then, after it reaches completion, the resulting mixture containing the products is added to the tested microorganisms. The second approach is based on running the reaction in situ, already in the presence of tested microorganisms [29]. In the present study, the second approach was applied because the physiological LPO system acts in situ and, in addition, there are reports showing a stronger antimicrobial effect for the LPO system acting in situ than the system acting ex situ [13]. This may be related to the formation of unstable and reactive intermediates of the LPO system, which are degraded during ex situ incubation.

In the present study, the assessment of the biofilm-related factors directly related to the development of dental caries in vivo (total biomass, extracellular polysaccharide mass, lactic acid synthesis, and glucose/sucrose consumption) was carried out in the presence of three H_2_O_2_ concentrations to reveal the relationship between the observed effect and the amount of LPO product generated. These preliminary results will be helpful in planning and carrying out future research on the antibiofilm potential of the modified LPO system. In this study, two experiments were also performed to explain the mechanisms of the observed changes. The first focused on the evaluation of PTS activity, which provided information on whether the detected alterations in these parameters were related to the impairment of intracellular carbohydrate transport. The second experiment, the assay of intracellular NAD^+^ and NADH concentrations, allowed the easy detection of metabolic changes at the level of intracellular carbohydrate metabolism. In both experiments mentioned above, the concentration of H_2_O_2_ was set at 1000 μM since it had the greatest effect on the other tested parameters.

In each of the experiments, PBS solution at pH 7.4 was used as the control sample to which all results obtained from the tested samples were compared. Furthermore, additional substrate controls—(pseudo)halides in PBS and H_2_O_2_ in PBS—were used to exclude the possibility that the observed effect was substrate-dependent. In the case of (pseudo)halides, no statistically significant changes compared to the control were observed in any of the experiments. In samples containing only H_2_O_2_, a small but statistically significant effect on biofilm mass was observed. Nevertheless, it should be noted that the observed effect of the LPO systems was associated with the production of a reactive LPO system product and not with H_2_O_2_ toxicity. The reason for this was the fact that the reaction mixture combined a small amount of H_2_O_2_ with high LPO activity, resulting in immediate consumption of all H_2_O_2_ during the halogenation cycle. The depletion of H_2_O_2_ occurred in the first few seconds after the start of the reaction, which is related to the extremely fast reaction of LPO compound I with (pseudo)halides [31].

Surprisingly, the inhibitory effect of the physiological LPO–SCN^−^ system was rather weak, despite its well-documented bacteriostatic and bactericidal effects on planktonic microorganisms [13,32,33,34]. In the present study, despite its inhibitory effect on PTS, only a small but statistically significant inhibitory effect on lactate synthesis was observed exclusively in 24 h biofilms. Interestingly, no inhibition of lactate synthesis was observed after incubation with the LPO system in 2 h biofilms, which, unlike 24 h biofilms, have an underdeveloped extracellular matrix. It should be noted that the products of the LPO–SCN^−^ system were hypothiocyanite ion or its protonated form hypothiocyanous acid. In fact, an acid–base equilibrium in these products was observed [35]. Lowering the environmental pH shifted the equilibrium towards the acidic (protonated) form. Since undissociated acidic forms penetrate biological membranes more easily than dissociated forms, high pH does not favour the penetration of dissociated reactive products from LPO systems into microbial cells, thus reducing their effectiveness [36]. Although the reaction itself was carried out in a pH 7.4 environment (maintained by PBS solution), local areas of lower pH may have occurred within the extracellular matrix and bacterial cells, especially in the well-developed matrixes of the 24 h biofilms, where protonation of the LPO product may have occurred, facilitating its penetration into bacterial cells. This may explain the greater inhibitory effect on lactate production in older biofilms. Considering the fact that the present study is pioneering in its assessment of the effects of LPO systems on biofilms in vitro, it was not possible to compare the obtained results to previous studies. However, there are results from clinical trials in which patients received preparations containing the glucose oxidase–LPO–SCN^−^ system alone [37] or in combination with other biomolecules, such as lactoferrin [38,39,40,41,42] and lysozyme [43]. Various parameters were analysed, including volatile sulphur compounds in exhaled air [41,44], colony forming units (CFUs) of *S. mutans* and *L. acidophilus* in saliva [39], total bacteria counts [39,43], oral health parameters [39], and the severity of xerostomia [40]. The results of these studies do not conclusively indicate a beneficial effect from preparations enriched with an LPO–SCN^−^ system. Although the studies conducted by Gudipaneni et al. [39] and Pinherio et al. [43] pointed to a significant decrease in the amount of *S. mutans* and the total amount of microorganisms in the biological material, the studies by Shimizu et al. [38] and Shin et al. [41] did not show a statistically significant effect. The differences between the results may have been due to the different compositions that the patients received, both in terms of concentrations as well as the addition of small biomolecules beyond the LPO system.

In contrast to the LPO–SCN^−^ system, the LPO–I^−^ system showed the greatest inhibitory effect on total biomass growth and insoluble polysaccharide mass and, importantly, caused complete inhibition of lactic acid synthesis. These effects may have been linked to the observed inhibition of carbohydrate transport into *S. mutans* cells and impairment of cell metabolism, as manifested by changes in intracellular NAD^+^/NADH concentrations.

The observed decrease in NADH concentration may have been related to the direct oxidation of this molecule by reactive iodide species synthesised by LPO. The products of the lactoperoxidase system generated from thiocyanate, iodide, and bromide have been shown to directly oxidise NADH and NADPH molecules [45], which is one of the nonspecific defence mechanisms against the toxicity of the LPO system in both host and microbial cells. The lack of intracellular NADH may have been one direct reason for the observed total inhibition of lactate synthesis in biofilms treated with the LPO–I^−^ system due to the role of NADH as a substrate for lactate dehydrogenase (LDH) [46]. Importantly, unlike the LPO–SCN^−^ system, the oxidation products of the LPO–I^−^ system are not NAD^+^ but a number of different degradation products (irreversible loss of the molecule) [47]. Therefore, the simultaneously observed increase in the concentration of NAD^+^ must be related to another mechanism where its increased synthesis takes place. Although the results of studies on LPO systems affecting the process of de novo NAD^+^/NADH synthesis are not available, there are studies investigating the effect of the system on the glycolysis process. Glyceraldehyde-3-phosphate dehydrogenase is an enzyme responsible for the conversion of glyceraldehyde 3-phosphate to 1,3-bisphosphoglycerate, with concomitant reduction of NAD^+^ to NADH. Reactive products of the LPO system have been shown to be capable of inhibiting this enzyme, thus stopping the reduction of NAD^+^ to NADH, which may explain the observed rise in its concentration due to accumulation [48]. Another mechanism explaining the increment in the NAD^+^/NADH ratio may be an increase in the expression of water-forming NADH oxidase in response to oxidative stress. This enzyme plays a key role in *S. mutans* cells by keeping NAD^+^ available for glycolysis and protecting it from oxidative stress [49].

The LPO–SeCN^−^ system can be considered an analogue of the physiological LPO system due to the substrate and product similarity between them. However, unlike LPO–SCN^−^, LPO–SeCN^−^ showed a greater ability to inhibit biomass growth and extracellular polysaccharide synthesis. Similarly to the physiological system, the inhibition of lactate synthesis was greater in 24 h biofilms, which may have also been related to the protonation of the generated product (OSeCN^−^) in a region of locally lower pH within the biofilm matrix. In contrast to the LPO–I^−^ system, an increase in the NAD^+^/NADH ratio was observed but so was a rise in the intracellular NADH concentration. It can be hypothesised that this was the reason for the inhibitory effect of the product of the LPO system on lactate dehydrogenase (NADH accumulation). To this can be added the inability of hyposelenocyanite to reduce the accumulated NADH to NAD or other degradation products, as in the case of the LPO–I^−^ system. However, this mechanism requires further in-depth verification, which is beyond the scope of this study.

The last system tested—LPO–SCN^−^+I^−^—showed the least antibiofilm properties and the lowest inhibition of carbohydrate metabolism of all the systems presented. The product of this system has been identified as cyanogen iodide (ICN) by Schlorke et al. [25]. The available data on the microbial effects of this product are insufficient. To the best of our knowledge, only one paper (by Schlorke et al. [17]) has demonstrated the effective bactericidal activity of this modification of the LPO system against *Escherichia coli* in planktonic culture. In our study, despite having similar working conditions (50 nmol/L LPO, SCN^−^:I^−^ ratio of 1:2, pH = 7.4), we did not observe a significant effect from the LPO–SCN^−^+I^−^ system on *S. mutans* biofilms. This may have been a result of increased resistance of *S. mutans* to ICN or impaired penetration of this product into the bacterial cells. This demonstrates the need for further research on this system. The focus should be on susceptibility testing of individual clinically relevant microbial species and studies of the stability of this product (possible degradation prior to biofilm penetration) and its ability to penetrate biological barriers, including the biofilm matrix.

All scientific studies have certain limitations. The main limitation of the present study was that it was carried out in vitro, which certainly affected the interpretation of the results obtained. In the environments of living organisms—more specifically, in the oral cavity—the number of factors influencing the action of the LPO system is much greater than under in vitro conditions. First of all, in a living organism, there are a large number of defence mechanisms that can inactivate reactive products of the LPO system. These include substances that affect the antioxidant capacity of saliva (small-molecule substances and proteins). Second, there is constant removal of saliva due to its continuous production in the oral cavity environment. It should be noted that, in the case of exogenous delivery of the LPO system without the use of carriers that would slow its release, the effect of the system would be limited to only a few minutes following application [50].

An important factor that also influences the action of the LPO system in vivo is the method of H_2_O_2_ delivery, the availability of which is a limiting factor for the lactoperoxidase reaction. In the present study, this substrate was delivered in one portion at the beginning of the experiment in a rapid initial synthesis during the first few seconds of the reaction and the sudden rapid depletion of the substrate. Furthermore, subsequent reactions between the generated products and the available H_2_O_2_ would not be unlikely, especially in the case of the LPO–I^−^ system [12]. H_2_O_2_ is continuously produced in vivo by host oxidases and microorganisms but in small amounts that are immediately scavenged by reactive oxygen species (ROS) defence enzymes and by peroxidases (including LPO) [19].

The pH of the oral environment also plays an important role in influencing the biological effect of the LPO system in vivo. Some products of the tested systems (hypothiocyanite, hyposelenocyanite, some of the reactive iodide species) are acidic substances that can undergo electrolytic dissociation, as described in previous work [35]. Furthermore, the LPO–I^−^ system generated different products depending on the environmental pH. These products can be characterised with different potential antibiofilm properties. In an environment with a pH below 6, the only reactive product is I_2_. At pH 6 to 9, the LPO–I^−^ system synthesises HOI, I_2_OH, I_2_, and I_3_^−^, with HOI being the most reactive in this environment. It is worth noting that oral hygiene products are alkaline (pH 7–10), which is important for the future development of LPO system-enriched products [51]. In view of the above, the effects of the LPO–SCN^−^ and LPO–SeCN^−^ systems in this type of preparation will be weakened by the dissociation of the main product in an alkaline environment. The opposite effect would be observed for the LPO–I^−^ system. In an alkaline environment, the predominant product is HOI, which is characterised by a strong reactivity toward thiol groups, thioether groups, amine moieties, and oxidation of NAD(P)H [47,52]. This is another favourable argument for further basic and applied research on the LPO–I^−^ system.

Further investigation of the use of a modified LPO system in the treatment of biofilm-related diseases should include optimising the concentration of each component, particularly H_2_O_2_, which is a limiting substrate in the reaction catalysed by LPO. A particularly promising approach for in vivo application appears to be the replacement of H_2_O_2_ in substantia with an H_2_O-generating system, such as glucose oxidase or lactate oxidase, in future oral hygiene products, but in this case, the selection of the enzyme that will be active in an alkaline environment is crucial. This would extend the period of efficacy for the LPO system by increasing the time that H_2_O is available and, more generally, solve the problem of the low stability of H_2_O in solutions, allowing a preparation with greater stability and a longer shelf life to be produced.

Finally, a necessary step will be to determine the lowest concentration of LPO system components that exhibit significant in vitro and in vivo antibiofilm effects while remaining nontoxic to host cells. In our previous studies on the toxicity of the described modifications in an LPO system against human gingival fibroblasts, we found that the LPO–I^−^ system was characterised by the highest toxicity and the physiological system (LPO–SCN^−^) by the lowest toxicity [29]. The results showed that modifying the LPO system enhances both its antibiofilm activity and its negative effects on host cells.

Much remains to be done to counteract cariogenic biofilms, as dental caries remains a major clinical, social, and financial problem. We are confident that the results presented add value and demonstrate the potential of the modified LPO system in controlling cariogenic biofilms and, in a broader perspective, biofilms associated with various other diseases.

## 4. Materials and Methods

### 4.1. LPO System Setup

In each of the experiments, the LPO concentration was 50 nmol/L, which reflected its concentration in human saliva. Each experiment was preceded by the preparation of an LPO solution, where 1 mg of the enzyme was dissolved in 1 mL of PBS, followed by measurement of absorbance at 412 nm (ε_412_ = 112,000) to determine the concentration.

(Pseudo)halide substrates were added in a concentration of 10 mM, which is about two times higher than that of the physiological LPO substrate (SCN^−^) present in human saliva [15]. This decision was based on the need to use the excess of the substrate over hydrogen peroxide (tenfold excess over the highest tested concentration of H_2_O_2_).

Three concentrations of hydrogen peroxide (1000, 500, and 250 μM) were used in the present study. The deficiency of H_2_O_2_ on the (pseudo)halide substrate played a limiting role in the reaction and additionally controlled the amount of product generated, which was equal in molar amount to the H_2_O_2_ provided.

The control sample that was used for comparison in the statistical analysis consisted of PBS without any additional LPO system components. To exclude a toxic effect from the substrates alone, additional samples containing only (pseudo)halide and only H_2_O_2_ in PBS were also included.

### 4.2. Biofilm Growth

The present study was conducted with the *S. mutans* ATCC 25,175 bacterial strain. It was grown in brain heart infusion (BHI) medium (Graso Biotech, Owidz, Poland) for 12 h, after which it was centrifuged (1200 RPM; 5 min). The remaining bacterial mass was then suspended in 5 mL of BHI and immediately used for the preparation of a suspension of 0.2 McF BHI + 5% sucrose. The suspension was added to 24-well plates (1 mL per well) or 96-well plates (250 μL per well), depending on the experiment. The evaluations of the selected parameters were performed with biofilms in the early biofilm growth phase (2 h) and 24 h old biofilms, depending on the experiment [53].

### 4.3. Total Biofilm Biomass Assay

The total biomass of the biofilm was assessed with the previously described crystal violet method [53]. First, the early (2 h) biofilm was placed in 96-well plates, as described in Section 4.1. After 2 h, the planktonic bacteria were isolated by carefully removing the growth medium. A fresh portion of 0.2 mL of PBS was then added, followed by the addition of the specific components of the LPO system. The enzymatic reaction was started by adding H_2_O_2_. After 15 min of incubation within this system, the solution above the biofilm was removed and replaced with a fresh portion of BHI + 5% sucrose. The biofilms were then incubated for 350 min at 37 °C in an atmosphere containing 5% CO_2_. Then, the planktonic bacteria were removed and the biofilms were stained with 0.1% crystal violet solution for 20 min, after which the excess dye was washed with PBS. The bound dye was extracted with a 33% acetic acid water solution and transferred to a fresh 96-well plate to measure the absorbance at 540 nm. The amount of crystal violet bound by the biofilm (absorbance) was directly proportional to the total biomass of the biofilm.

### 4.4. Insoluble Extracellular Polysaccharide Mass Assay

Insoluble extracellular polysaccharide biomass was measured using the anthrone–sulphuric acid method [54]. The biofilms were prepared, treated with LPO systems, and incubated in the same way as described in Section 4.2. After incubation, the biofilms were washed with PBS and scraped from the bottom of the wells with pipette tips. The biofilm released was then transferred to an Eppendorf-type tube and centrifuged (5500 RCF; 5 min). After the supernatant liquid was removed, the biofilm was washed with PBS and centrifuged three more times. Once the soluble saccharides had been removed from the biofilm matrix, the water-insoluble fraction was dissolved by adding 500 µL of 1 M NaOH, which was followed by incubation at room temperature for 30 min with constant mechanical stirring. Next, the insoluble microorganic and biofilm residue was centrifuged and 20 μL of each supernatant solution was transferred to a 96-well plate. The mass of the extracted polysaccharide was evaluated using the anthrone method by adding 200 μL of 0.15% anthrone solution in 80% (*v*/*v*) H_2_SO_4_. The absorbance at 626 nm was measured after 30 min of incubation at 95 °C. The absorbance was directly proportional to the mass of insoluble polysaccharide found in the biofilm.

### 4.5. Lactate, Glucose, and Sucrose Concentration Assay

Lactate and glucose concentrations were measured in biofilm growth medium with a direct bioamperometric method utilising the Biosen C-line Analyzer (EKF Diagnostics), which allows automatic measurement of lactate and glucose concentrations in the range of 0.5–40 mM. The device was equipped with lactate and glucose enzymatic biosensors for the simultaneous assaying of the glucose and lactate concentrations from one sample.

First, 2 and 24 h biofilms in 24-well plates were subjected to the tested LPO systems for 15 min and then gently washed three times with 1 mL of PBS, after which 1 mL of BHI with 1% of sucrose was added to the wells. Lactate, glucose, and sucrose concentrations were measured every 50 min for 350 min (with the first measurement taking place immediately after adding the medium). The measurement was performed by transferring 10 μL of growth medium from each well to 500 μL of System Solution (EKF Diagnostics, Cardiff, United Kingdom) in an Eppendorf-type tube. The tubes were inserted into the analyser, and the measurement was performed. The sucrose concentration was determined by transferring 20 μL of the sample to a 50 μL solution of invertase from *Sacharomyces cerevisiae* (Sigma-Aldrich, Schnelldorf, Germany) (1 mg/mL in 0.05 M citrate buffer, pH = 4.5). The samples were incubated at 55 °C for 20 min, which ensured complete sucrose hydrolysis. This was followed by measurement of the glucose concentration in the digested sample with the same method as described previously (sum of glucose from BHI and glucose after breakdown). The sucrose concentration in each sample was calculated by subtracting the glucose concentration before hydrolysis from the total glucose concentration (the number of moles of glucose generated from sucrose hydrolysis was equal to the number of moles of sucrose present in the sample). The results of the assay were then calculated, applying the correction for volume changes (decrease in total medium volume in wells after each measurement), and expressed as lactate/glucose/sucrose moles for the well.

### 4.6. PTS System Activity Assay

The overnight culture of *S. mutans* in BHI medium was put in falcon tubes and then centrifuged and washed with PBS. To obtain log phase cells, a fresh portion of BHI medium was added and the bacteria were growth for 2 h. After that, the cells were again centrifuged and washed with PBS. The suspension of 2 McF was prepared in PBS and 1 mL of that suspension was transferred to Eppendorf-type tubes. To each tube, the components of the LPO system were added (full system—LPO + (pseudo)halides + hydrogen peroxide (1000 μmol/L); control samples—only PBS, only (pseudo)halides); the control sample consisting of the suspension of *S. mutans* without any additions). After 15 min of incubation, the tubes were centrifuged and the supernatant was decanted. The cells were suspended in PBS with 5 mM Mg^2+^ (decryptification buffer) and permeabilized with 50 µL of acetone–toluene mixture (9: 1) for 5 min. The activity of PTS was measured based on the method described by LeBlanc et al. [55]. The reagent mixture consisted of 10 mM glucose, 10 mM sodium fluoride, 0.1 mM NADH, and 10 U/mL lactic acid dehydrogenase in PBS with 5mM Mg^2+^. Measurements were carried out in 96-well plates by adding 75 μL of *S. mutans* suspension and 200 μL of reagent mixture to each well. The reaction was started by adding phosphoenolpuryvate (PEP) (final concentration: 5 mM), and the change in NADH concentration was measured for 4 min by detecting the change in absorbance at 340 nm. Simultaneously, the second sample without the addition of PEP was analysed to correct the result (subtraction of NADH oxidation without the dependence of PTS activity). Finally, the result was expressed as nanomoles of oxidised NADH per second depending on PEP consumption by the PTS system [49].

### 4.7. NAD^+^/NADH EC Assay

The influence of the LPO systems on biofilm NAD^+^ and NADH levels was evaluated by sweeping using borate complexation capillary electrophoresis. This approach allowed us to measure NAD^+^ and NADH at very low concentrations (0.5 μg per well/single biofilm). In the present study, a modified version of the method described by Rageh et al. [44], originally developed for a urine nucleotides assay, was utilized.

After 15 min of treatment with the tested LPO systems, the 24 h biofilms were washed three times with PBS and once with molecular-grade water. Next, the supernatant liquid was discarded and the biofilms that remained in the 24-well plate were frozen at −20 °C and then lyophilised for 4 h to eliminate all the residual water from the samples. In each well, 600 μL of ice cold 1:1 acetonitrile–water mixture was added, and then the biofilm was scraped using a pipette tip. All the released biofilm biomass was transferred to a 1.5 mL tube and 0.2 g of glass beads was added. After that, the extraction of NAD^+^/NADH was performed for 60 min with 30 s of shaking every 5 min. During extraction, the samples were incubated at 4 °C. Finally, the samples were centrifuged at 14,000 RCF for 5 min at 4 °C and then the supernatant was collected in a fresh 1.5 mL tube.

The concentrations of NAD^+^ and NADH in the extracts were determined with capillary electrophoresis using a sweeping borate complexation method. Separation was performed using the Prince Technologies CE System equipped with the Bischoff Lambda 1010 UV/VIS detector set to 254 nm. All separations were performed using fused-silica capillaries, 75 μm in internal diameter and 65 cm in length (effective length: 50 cm). Each day of the experiment, the capillary was rinsed with NaOH 1 M (5 min), water (5 min), and buffer (5 min). As a background electrolyte (BGE), 25 mM borate buffer (pH = 9.25) was used. The separation process was aided by the addition of 5 mM of the β-cyclodextrin (5 mM in BGE), which has been described by several authors to increase selectivity by increasing mobility differences for nucleotides with the same mass/charge ratio [44,56,57]. The samples, diluted three times with molecular-grade water, were injected into the capillary at a pressure of 200 mbar for 50 s. After that, the online sample concentration process was started by applying negative voltage (−30 kV) until the end of the sweeping process (until the current reached 90% of the current measured in the BGE-only filled capillary). Separation was started by applying 30 kV of positive voltage. After each measurement, the capillary was rinsed with 1 M NaOH, water, and BGE (1 min for each). Signal detection and qualitative and quantitative analyses were performed using DAx software v8.0 (Prince Technologies, Emmen, The Netherlands). The concentrations of NAD^+^ and NADH were calculated by comparing the obtained peak areas with the peak areas of the standard solutions.

### 4.8. Statistical Analysis

The R environment v4.3 (R Foundation for Statistical Computing) was utilised for data analysis. The ggplot2 package was used for plot preparation. Variances’ homogeneity was determined with Levene’s test. Statistical significances between groups were evaluated using one-way analysis of variance (ANOVA) with the post hoc Tukey test.

## 5. Conclusions

The lactoperoxidase system can be an effective agent of natural origin against cariogenic biofilms. This requires some modifications to change the physiological substrate to an alternative (pseudo)halide substrate, mainly selenocyanate and iodide. The effects of these changes are associated with impaired carbohydrate transport and metabolism, resulting in reduced expression of key virulence factors of *S. mutans*, particularly the ability to produce biofilms that include the polysaccharide matrix and lactic acid.

## Figures and Tables

**Figure 1 ijms-24-12136-f001:**
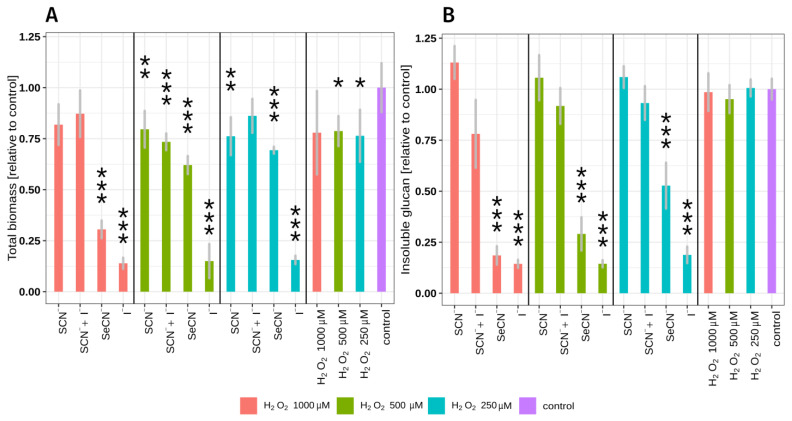
Influences of all tested LPO system modifications on *S. mutans* total biofilm biomass (**A**) and total insoluble polysaccharide mass (**B**). The (pseudo)halides on the *x*-axis refer to the substrates used in the tested LPO system. Values are expressed as fractions of control samples (which were equal to 1) to eliminate variations observed among different plates. Each colour represents a system with a given hydrogen peroxide concentration tested in the experiment. Biofilm treated only with PBS was used as the control sample. Asterisks indicate a statistically significant difference between a tested group and the control in the post hoc Tukey’s test—* *p* < 0.05; ** *p* < 0.01; *** *p* < 0.001.

**Figure 2 ijms-24-12136-f002:**
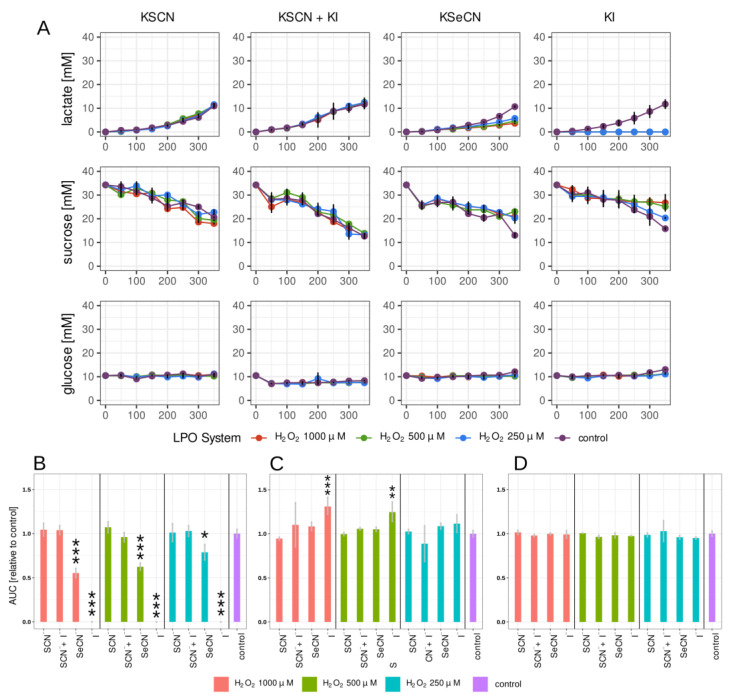
(**A**) Kinetic measurements of lactate production and sucrose and glucose consumption in 2 h biofilms treated for 15 min with tested LPO systems. The curves for each of the tested concentrations of H_2_O_2_ added to the systems are shown in different colours (legend above the graph). (**B**–**D**) Areas under kinetic curves (AUCs) (relative to control): (**B**) lactate; (**C**) sucrose; (**D**) glucose. The (pseudo)halides on the *x*-axis refer to the substrates used in the tested LPO system. For lactate, lower values indicate slower lactate production in the sample. In the case of glucose/sucrose, higher values indicate reduced consumption of glucose/sucrose by the biofilm. Different concentrations of H_2_O_2_ in the tested systems are represented by different colours (see legend). Biofilm treated only with PBS was used as the control sample. Asterisks indicate statistically significant differences between tests. * *p* < 0.05; ** *p* < 0.01; *** *p* < 0.001.

**Figure 3 ijms-24-12136-f003:**
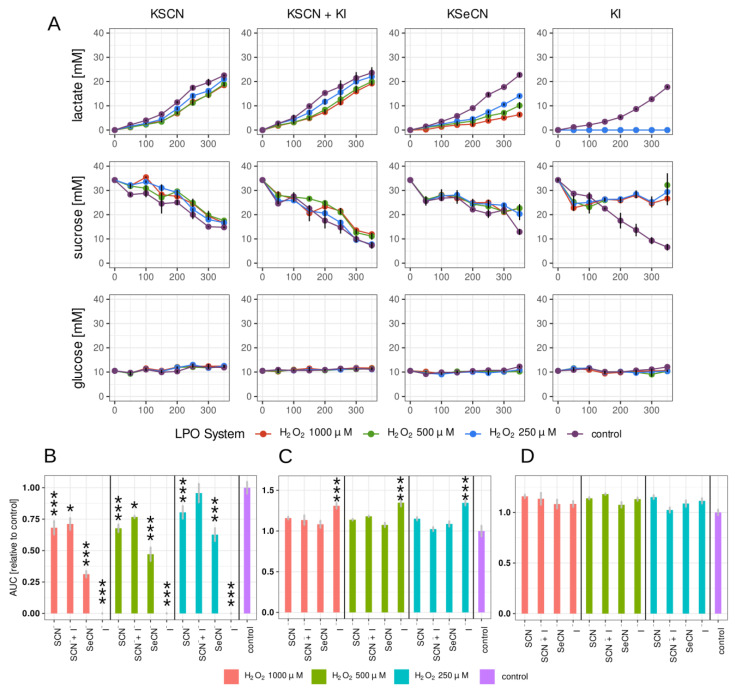
(**A**) Kinetic measurements of lactate production and sucrose and glucose consumption in 24 h biofilms treated for 15 min with tested LPO systems. The curves for each of the tested concentrations of H_2_O_2_ added to the systems are shown in different colours (legend above the graph). (**B**–**D**) Areas under kinetic curves (AUCs) (relative to control): (**B**) lactate; (**C**) sucrose; (**D**) glucose. The (pseudo)halides on the *x*-axis refer to the substrate used in the tested LPO system. For lactate, lower values indicate slower lactate production in the sample. In the case of glucose/sucrose, higher values indicate reduced consumption of glucose/sucrose by the biofilm. Different concentrations of H_2_O_2_ in the tested systems are represented by different colours (see legend). Biofilm treated with PBS only was used as the control sample. Asterisks indicate statistically significant differences between a tested group and the control in the post hoc Tukey’s test—* *p* < 0.05; *** *p* < 0.001.

**Figure 4 ijms-24-12136-f004:**
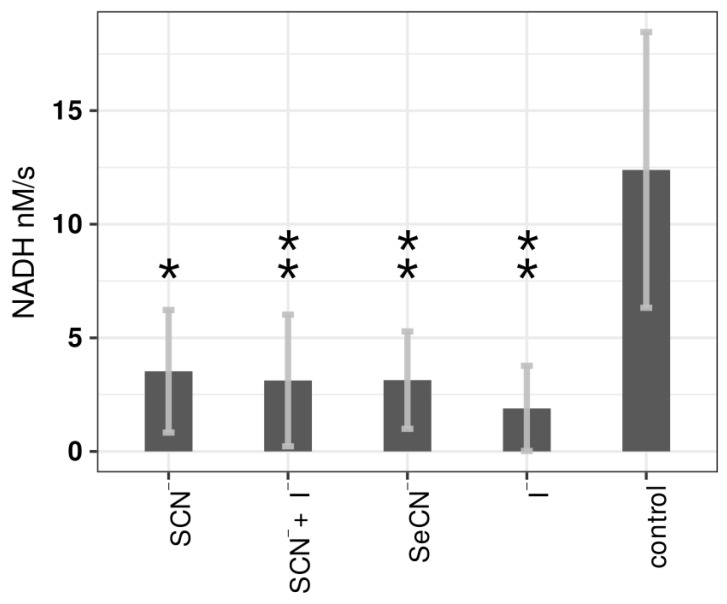
PTS system activity in *S. mutans* cells treated with tested LPO system modifications and control sample. The (pseudo)halides on the *x*-axis refer to the substrates used in the tested LPO systems. Biofilm treated with PBS only was used as the control sample. Asterisks indicate a statistically significant difference between a tested group and the control in the post hoc Tukey’s test—* *p* < 0.05, ** *p* < 0.01.

**Figure 5 ijms-24-12136-f005:**
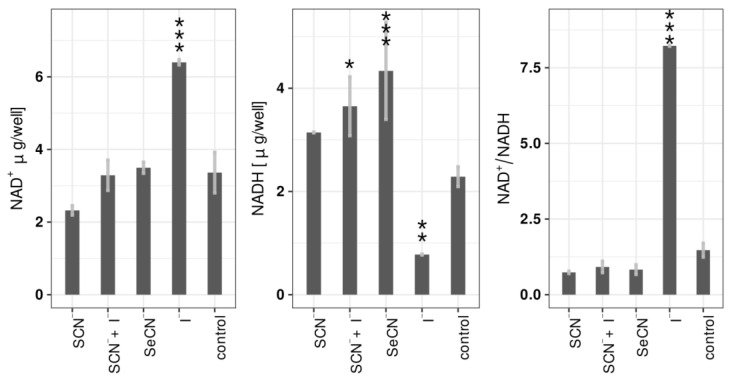
Amounts of NAD^+^ and NADH in the well and NAD+/NADH ratio after treating the biofilm with all tested systems (1000 μM H_2_O_2_). The (pseudo)halides on the *x*-axis refer to the substrates used in the tested LPO systems. Asterisks indicate a statistically significant difference between a tested group and the control in the post hoc Tukey’s test—* *p* < 0.05, ** *p* < 0.01; *** *p* < 0.001.

## Data Availability

Not applicable.

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
