# Peer review of "Modified Lactoperoxidase System as a Promising Anticaries Agent: In Vitro Studies on Streptococcus mutans Biofilms"

_ijms, 2023, doi:10.3390/ijms241512136_

Round 1
Reviewer 1 Report
Overall, this is an excellent study, but the reviewer has questions and comments regarding some points in the manuscript.
Section "Introduction"
It seems that this section requires revision, since after reading it does not become clear why the authors conducted this study. There are a large number of studies that describe the role of lactoperoxidase in maintaining oral health (see, for example, the review article https://www.mdpi.com/1422-0067/20/6/1443, which was written by the authors of this manuscript). Moreover, there are a significant number of commercial products containing the lactoperoxidase system in combination with certain additional components, but the authors do not write about it.
The reviewer believes that in this section the reader's attention should be focused on such points as: i) Microorganisms in biofilms can be difficult to reach for biocides (products of the lactoperoxidase system) and ii) This can be combated by increasing the concentration of biocides by increasing the concentrations of their precursors. Then it will become clear why in this study iodide, for example, or hydrogen peroxide, and why the concentrations of these substances for the study were used higher than physiological ones. Are there halides in saliva? And after all, some oral streptococci are capable of producing hydrogen peroxide? In addition, the human population is variable in the content of lactoperoxidase in saliva, which is influenced by many factors, including genetic ones.
It seems that the readers should be explained in more detail why, in order to assess the effects of the lactoperoxidase system on streptococci and their biofilms, the authors chose carbohydrate transport systems and the NAD+/NADH ratio.
Lines 99-102, it seems, here the authors need to reformulate their statement, since its meaning is unclear.
Section "Materials and Methods".
Line 486 (2.1. Biofilm growth). It looks like the numbering of the subsections is broken here.
Line 552. The ratio (9:1) of acetone-toluene mixture is repeated twice in one line. This is probably a mistake.
The authors should explain why for the PTS system activity assay only lactoperoxidase controls or only halides were used as controls; in other experiments described in the manuscript, the reviewer did not find such controls. At least, it seems that in the rest of the experiments, only PBS was used as a control. Why is there no hydrogen peroxide as a control? Obviously, for example, if there were such controls in the "NAD+/NADH EC assay" subsection, then many questions would not arise after reading the "Results and Discussion".
Author Response
Dear Reviewer,
we are grateful for a positive evaluation of our manuscript and for the helpful comments. We took a positive attitude to all of them and made the suggested corrections in the text. We enclose the revised version of the manuscript.
We hope that in the present form, we have met the expectations of the reviewers and it will be possible to publish the manuscript in the International Journal of Molecular Sciences
Overall, this is an excellent study, but the reviewer has questions and comments regarding some points in the manuscript.
It seems that this section requires revision since after reading it does not become clear why the authors conducted this study. There are a large number of studies that describe the role of lactoperoxidase in maintaining oral health (see, for example, the review article https://www.mdpi.com/1422-0067/20/6/1443, which was written by the authors of this manuscript). Moreover, there are a significant number of commercial products containing the lactoperoxidase system in combination with certain additional components, but the authors do not write about it.
Authors reply: Information on the availability of oral hygiene products containing lactoperoxidase and other biomolecules on the market has been added to the Introduction. This aspect is also addressed in the Discussion (in the revised version, this section has been separated from the Results). We have also highlighted the fact that such preparations contain only thiocyanate as a substrate for lactoperoxidase, whereas other substrates for this enzyme have been described and their oxidation products may potentially have stronger antibiofilm properties. The introduction section has been extensively rewritten to more clearly outline the problem and indicate the purpose of the study.
The reviewer believes that in this section the reader's attention should be focused on such points as i) Microorganisms in biofilms can be difficult to reach for biocides (products of the lactoperoxidase system) and ii) This can be combated by increasing the concentration of biocides by increasing the concentrations of their precursors. Then it will become clear why in this study iodide, for example, or hydrogen peroxide, and why the concentrations of these substances for the study were used higher than physiological ones. Are there halides in saliva? And after all, some oral streptococci are capable of producing hydrogen peroxide. In addition, the human population is variable in the content of lactoperoxidase in saliva, which is influenced by many factors, including genetic ones.
Authors reply: The Introduction section has been rewritten to follow the suggested format. We have added the paragraph describing the stages of biofilm formation, highlighting the fact that as the matrix is synthesized and matures, the microorganisms within it become less exposed to be reached by biocides. We have also highlighted the fact that only thiocyanates can be found in significant concentrations in saliva, making them the main substrate for endogenous lactoperoxidase. We have also briefly described the source of hydrogen peroxide and how it is delivered in commercially available oral hygiene products.
It seems that the readers should be explained in more detail why, in order to assess the effects of the lactoperoxidase system on streptococci and their biofilms, the authors chose carbohydrate transport systems and the NAD+/NADH ratio.
Authors reply: We have added a discussion regarding the rationale for determining PTS activity and the NAD+/NADH ratio (5th paragraph of Discussion). It should be noted that the presented study in the field of mechanism research can be qualified as a preliminary study for further metabolomics research, which is beyond the scope of the present study and is planned by the authors in the near future. The data obtained from the two experiments made it possible to evaluate the activity of the PTS and to obtain information on whether the changes observed in these parameters were related to the impairment of intracellular carbohydrate transport. The determination of intracellular NAD+ and NADH concentrations allowed easy detection of metabolic changes at the level of intracellular carbohydrate metabolism.
In lines 99-102, it seems, here the authors need to reformulate their statement since its meaning is unclear.
Authors reply: Thank you for pointing out the unclear paragraph. We have revised the whole paragraph to make the aim of the study clearer.
Line 486 (2.1. Biofilm growth). It looks like the numbering of the subsections is broken here.
Authors reply: The numbering has been corrected where indicated. Due to the separating of the Results and Discussion into two sections, we have also reviewed the numbering throughout the text.
Line 552. The ratio (9:1) of the acetone-toluene mixture is repeated twice in one line. This is probably a mistake.
Authors reply: We have removed the repetition.
The authors should explain why for the PTS system activity assay only lactoperoxidase controls or only halides were used as controls; in other experiments described in the manuscript, the reviewer did not find such controls. At least, it seems that in the rest of the experiments, only PBS was used as a control. Why is there no hydrogen peroxide as a control? Obviously, for example, if there were such controls in the "NAD+/NADH EC assay" subsection, then many questions would not arise after reading the "Results and Discussion".
Authors reply: Thank you for pointing out this aspect. In the manuscript, the type of control used was incorrectly stated. In the PTS activity analysis, as in other experiments, the control was a PBS solution without LPO system components. We have also added to the discussion the description of the control sample and the samples containing substrates only. In Table S5, we have added the results of the PTS activity assay from H2O2 treatment only (no statistically significant difference), which were not previously shown.
Thank You in advance for Your time and very reliable assessment.
Sincere regards,
Authors

Reviewer 2 Report
This manuscript presents a novel topic, and the authors have done a good job. However, there are small things that need careful improvement, and the manuscript has the merit to be published after revision. The following points should be addressed:
1. The English language requires further revision before reaching the final decision.
2. Express "in vitro" in italics in the title and in line 92.
3. Line 30: The sentence containing the full name and acronym "phosphoenolpuryvate PEP-carbohydrate phosphotransferase (PTS)" should be reformulated for better clarity.
4. For the paragraph (lines 60-63), the referee suggests including the article [Novel Options to Counteract Oral Biofilm Formation: In Vitro Evidence. Int J Environ Res Public Health. 2022 Jun 30;19(13):8056. doi: 10.3390/ijerph19138056] to enhance the quality of the manuscript.
5. Line 76: Improve the sentence where the expression "they can also can enter..." is used.
6. In line 162, delete the dot at the end of the subtitle.
7. Correct the word "...-iodidte" in line 385.
8. Please improve the expression "...in the field of combating..." as it doesn't fit well in this context.
9. The referee noticed the use of "...M" as a symbol, but the SI symbol is "nm"; please ensure consistent usage of symbols throughout the text.
10. Avoid using the full name of S. mutans after its acronym in subsequent mentions (lines 487 and 602).
Dear Editor,
thank you for providing me with the opportunity to participate in the review process.
The authors have made significant contributions to the manuscript, but kindly is requested a minor revision to address certain points, particularly regarding the English language used in various sections.
Best regards!
Author Response
Dear Reviewer,
we are grateful for a positive evaluation of our manuscript and for the helpful comments. We took a positive attitude to all of them and made the suggested corrections in the text. We enclose the revised version of the manuscript.
We hope that in the present form, we have met the expectations of the reviewers and it will be possible to publish the manuscript in the International Journal of Molecular Sciences
Below are the responses to the submitted comments:
Ad 1. We have checked the text one more time and corrected any remaining language errors. The changes have been highlighted.
Ad 2. We changed the text typeface to italics
Ad 3. We have rewritten the sentence. We have shortened it and listed the parameters more clearly.
Ad 4. Thank you for suggesting the article. We have referenced it in the place mentioned.
Ad 5. We have removed the repeated word and checked the text for linguistic accuracy.
Ad 6. We have removed the unnecessary dot.
Ad 7. We have corrected the spelling of the mentioned word.
Ad 8. We have changed the sentence by removing the unnecessary phrase.
Ad 9. We have revised the spelling of the units. To avoid confusion, we have changed the way concentrations are expressed from nM to nmol/l.
Ad 10. We have abbreviated the spelling where indicated and reviewed the entire text for unnecessary full names.
Thank You in advance for Your time and very reliable assessment.
Sincere regards,
Authors

Reviewer 3 Report
The manuscript by Magacz et al. investigates the in vitro effect of different LPO systems on S. mutans biofilm and sugar metabolism.
The article provides new insights into the effect of different LPO systems on biofilm development, lactic acid production, and sucrose metabolism.
Although the study was conducted in-vitro I believe that it's a good basis for further study.
However, I have a few concerns and remarks:
General:
The authors used 4 different LPO systems which are sometimes referred to by the acronym and sometimes by the long compound name. The mixture between the two in the article makes it hard to follow.
Introduction:
I think that a little bit of elaboration on biofilm formation (steps of initial attachment, maturation, and EPS production) is needed for readers that are less familiar with the subject, especially because the journal is with broad reader spectrum.
results and discussion:
1. in general, I think that this part should be we-write. The organization of the discussion, and results discussion, in each paragraph, makes it hard to follow and in some cases, it feels more like a review pepper than a discussion of a research article.
2. In all x-axis please delete the word "system" the figure will look less curded.
3. Figure 1- there are two colons with *, please add the one-star meaning in the figure legends. It is unclear which comparison was conducted for the statistic. Did the authors compare the control with no treatment at all to each treatment or did the H2O2 alone in each concentration serve as control?
4. Please change the Y axis in Figure 1 to have the number 1, it will be clearer.
5. Figure 2- please replace C and D so that it will appear in the same order as A and as in the text.
6. along the text please provide references when there is a description of the results. Many times, it appears in between the discussion it is hard to follow when it's the new results from the current study.
7. Biofilm of S. mutans at an early stage (2h) does not have a shield of EPS that might protect it from penetration of different treatments. Therefore, the difference can not be related to penetration to the biofilm (Line 268-269).
8. I can understand the interest in biofilm rather than planktonic bacterial suspension. However, it is still important to understand the bactericidal and bacteriostatic effects especially when the treatment was given at such initial stage of the biofilm formation. Please provide data on the live/dead ratio in the biofilm biomass. (Same can be done by confocal microscopy for the EPS to quantify their generation. This point is important since if there are fewer bacteria alive and also in some treatments the author observed a reduction in biofilm biomass it can explain the reduction in sucrose consumption and less lactic acid production. In general, I believe that this information should be normalized to living cells. Same for the PTS function and NADH/NAD.
9. change the second Figure 3 to Figure 4.
material and methods:
1. The material and method section has a few broken sentences. For example, line 479- After the planktonic bacteria (were removed?) same in line 506, After the supernatant liquid (was discarded?)
2. I didn't find when the experiment was performed on the initial biofilm stage (2h) and when on mature biofilm (24 h) I believe that this comparison is very important, and the author should include experiments and results on both conditions.
3. When the biofilm at an early stage was tested (2h) it is unclear whether the biofilm was washed and only the adherent bacteria were kept for further treatment with the compound.
4. Metabolism experiments it is better to perform in media without glucose (such as TY).
5. PTS system activity assay- please add the H2O2 concentration that was used also in the M&M.
references:
please go over the references and make sure they were cited in the right context (for example ref 61).
minor:
Line 192- in-vitro - put in italics.
Line 486- change 2.1 to 3.1
Line 549 change minu to min
Line 556- S. mutans - put in italics
Line 556- correct the u to μ
There are some broken sentences, missing the beginning or the end of the sentence.
Author Response
Dear Reviewer,
we are grateful for a positive evaluation of our manuscript and for the helpful comments. We took a positive attitude to all of them and made the suggested corrections in the text. We enclose the revised version of the manuscript.
We hope that in the present form, we have met the expectations of the reviewers and it will be possible to publish the manuscript in the International Journal of Molecular Sciences
Below are the responses to the submitted comments:
The authors used 4 different LPO systems which are sometimes referred to by the acronym and sometimes by the long compound name. The mixture between the two in the article makes it hard to follow.
Authors reply: Thank you for bringing this inaccuracy to our attention. We have decided to use the full names of the compounds in the text for the first use and the individual ionic formulae for subsequent uses. For example, for the LPO system with thiocyanate and iodide mixture, the abbreviation used is LPO-SCN-+I-.
I think that a little bit of elaboration on biofilm formation (steps of initial attachment, maturation, and EPS production) is needed for readers that are less familiar with the subject, especially because the journal is with broad reader spectrum.
Author's reply: The Introduction section has been expanded to include a description of the stages of biofilm formation. In addition, to provide a better introduction for readers less familiar with the subject, we have restructured this section and added a brief overview of currently available preparations containing LPO in their formula. Hopefully, these changes better outline the problem we are trying to address in this study.
Results and discussion.
ad 1.Thank you for bringing this to our attention. We have split the Results and Discussion sections into two separate sections. In the revised discussion, we first discuss the overall research conducted and the methods used. In the following section, we have successively discussed each of the systems examined in light of the results obtained. We hope that this approach will make it much easier to follow the results and the discussion for each of the systems investigated.
ad 2. We have removed the word "system" from all figures.
ad 3. We have added a one-star meaning to the diagram. Thank you for pointing out the ambiguity in the description of the control used. In all experiments, the control to which results obtained from samples consisting of the complete LPO system as well as its components ((pseudo)halide alone, hydrogen peroxide alone) were compared was the sample consisting of PBS alone. We have clarified this fact by adding the appropriate information in the figure description and in the 6th paragraph of the Discussion.
ad 4. We have changed the scale on the y-axis so that there is a marked value of 1. We have changed the scale so that it is identical to the adjacent graph.
ad 5. We have changed the order of the graphs as suggested. We have checked the figure references in the text.
ad 6. We have revised references throughout the text. We have removed unnecessary references and added others where necessary. In addition, we hope that the separation of the Results and Discussion sections will also improve the clarity of the description of the results obtained and the references to them.
ad 7. Thank you for this valuable comment. We have added to the manuscript additional results from the analysis of the effects of the studied systems on lactate synthesis and glucose and sucrose consumption in 2 h biofilms. This allowed the results to be analyzed in a broader context. We believe that the observed effect may be related to the lower level of acidification of the biofilm environment after 2 hours compared to older biofilms. In the case of the LPO-SCN system, the products formed are in equilibrium between the protonated and deprotonated forms. As pH increases, the equilibrium shifts towards the formation of the deprotonated form, which has poorer permeability across biological barriers, which may result in a weaker antimicrobial effect of the LPO system. This aspect is further discussed in point 7 of the Discussion section.
ad 8. We agree with the suggestion to perform additional studies using confocal microscopy. The present study is the first part of a larger, three-year study on the influence of lactoperoxidase systems on biofilms (study funded by Polish National Science Centre grant number 2021/41/N/NZ7/03315 (209,990 PLN)). The next planned stage of research will include studies of the influence of the described LPO systems on the structure of both single-species and multi-species cariogenic biofilms. The research will use confocal laser scanning microscopy. It is planned to evaluate parameters such as viability and distribution of microorganisms in the biofilm, extracellular polysaccharide distribution, and effects on species composition in mixed biofilms (FISH). As these studies are beyond the scope of this work, we have decided to focus here on a preliminary evaluation of the effects of the investigated systems on biofilm formation and carbohydrate metabolism (which, to our knowledge, has not been done before).
ad 9. We have corrected and re-checked the numbering of the figures.
Materials and methods
ad 1. We have reworked the entire text, improving broken sentences, typos, and text formatting.
ad 2. Thank you for pointing out this important fact. The 2h biofilms were used in the total biofilm biomass and extracellular polysaccharide mass assays. The 2h biofilms were treated with the tested systems for 15 min and then cultured for a further 350 min. We chose to perform these experiments on biofilms at the initial stage of formation because no further significant mass growth could be observed in mature biofilms (after 24h, biofilms reached their maximum mass under the applied conditions).
In the case of the lactate, glucose, and sucrose assays, we performed experiments on both 2 h and 24 h biofilms. In the revised version of the manuscript, following the reviewers' suggestions, we have added the results of the experiments on the 2 h biofilms.
The NAD+/NADH assay was performed only on 24 h biofilms. We decided to perform this assay only on 24 h biofilms because this experiment was designed to show the mechanism of action of each LPO system on 24 h biofilms that had reached a biomass large enough to allow determination of NAD+/NADH with acceptable accuracy and precision.
We have added the above information to the Materials and Methods (4.3, 4.4, 4.5), Results, and Discussion sections of the manuscript.
ad 3. Thank you for raising this important point. In the case of the 2 h biofilms, the washing was done very gently using an automatic pipette by a scientist experienced in biofilm cultivation. It is important to note that under the conditions used (BHI medium supplemented with a high concentration of sucrose - 5% and relatively dense initial bacterial suspension - 0.5 McF), adhesion and extracellular polysaccharide synthesis start very quickly. After 2 hours, the forming biofilm can be seen with the naked eye, making it possible to control the rinsing process so as not to damage the biofilm.
ad 4. We would like to thank you for this excellent advice, which we will take into account in our future research. In this research, we chose the BHI medium because of its high nutrient content for S. mutans (S. mutans has relatively high nutrient requirements). The use of this medium allowed us to simulate highly procariogenic conditions. This choice was also based on our team's extensive experience with biofilm cultures using BHI medium and the well-documented use of BHI medium in S. mutans experiments in the literature.
ad 5. We have added information about the hydrogen peroxide used (1000 µM) and a description of the reason for this choice.
References:
We have checked the references used and corrected them where necessary. In the case of reference 61 (revised version 50), it has been cited in the correct context - in relation to the PTS activity assessment method.
Minor: We have revised the entire text and corrected the mistakes indicated.
Thank You in advance for Your time and very reliable assessment.
Sincere regards,
Authors

Round 2
Reviewer 1 Report
The reviewer believes that the authors have fully answered his questions and comments.
The manuscript can be accepted for publication after making minor amendments, this applies to line 593; Is there an error in the concentration of hydrogen peroxide? The concentration indicated by the authors is 1000mol/l. It should probably be 1000 micromoles.
Author Response
Dear Reviewer,
we are grateful for a positive evaluation of our manuscript and for the helpful comments. We took a positive attitude to all of them, and made the suggested corrections in the text. We enclose the revised version of the manuscript.
Thank you for pointing out the minor amendments. There was indeed an error in the spelling of the unit as indicated, which we have corrected in the text.
We hope that in the present form we have met the expectations of the reviewers and it will be possible to publish the manuscript in International Journal of Molecular Sciences.

Reviewer 3 Report
I believe that the separation of the results and discussion was a good choice.
I still think that lactate secretion and metabolic state should be normalized to biofilm biomass (CFU/ CV).
I'm wondering why in the PTS and NAD/NADH experiments the control was PBS and not H2O2 at the same concentration as in the applied LPO system.
Minor comments:
Line 143- in vitro - italics
Line 209- I think the figure below hides the end of the figure legends.
Lines 338-340- "at the same time.... future" can be removed.
Author Response
Dear Reviewer,
we are grateful for a positive evaluation of our manuscript and for the helpful comments. We took a positive attitude to all of them, and made the suggested corrections in the text. We enclose the revised version of the manuscript.
We hope that in the present form we have met the expectations of the reviewers and it will be possible to publish the manuscript in International Journal of Molecular Sciences.
Ad. Minor comments:
Thank you for pointing out the minor errors, we have made all the necessary corrections indicated. The legend for the mentioned graph was placed above it, so in order to standardise the graphical presentation we have placed the legend below the graph in question and enlarged the legend slightly.
Reviewer: I still think that lactate secretion and metabolic state should be normalized to biofilm biomass (CFU/ CV).
Author’s reply:
We are very grateful for this comment. At the design stage of the experiment we considered normalising the results of the lactate, glucose and sucrose measurements to CV. However, we concluded that this may cause some difficulties in interpretation. This is related to the fact that, on the basis of our previous experiments, we observed that a 24 h biofilm cultured under such procariogenic conditions (high nutrient medium, high sucrose concentration, high initial suspension density of S. mutans) does not increase its biomass during the subsequent 350 min culture in fresh medium. However, a 2 h biofilm increased its total biomass. Instead, we used normalisation on a per-well basis (1 ml of medium at the start of the experiment). Regarding the standardisation with the CFU assay , based on our experience and previously published studies, we believe that the CFU assay of the biofilms we tested has a very high error rate. This is due to the production of a large amount of extracellular matrix by S. mutans, which makes it difficult to obtain a suspension of individual cells (despite methods involving mechanical mixing and sonication - high risk of cell killing), which is essential for a reliable CFU/biofilm assay. Taking this into account, we have decided in this experiment to express the result as the concentration of the substance in question per culture volume (1 ml of medium). In any case, we thank you for these comments and will certainly take them into account in our other studies, where the type of biofilm obtained and the way the experiment is carried out will allow the results to be presented in the discussed way.
Reviewer: I'm wondering why in the PTS and NAD/NADH experiments the control was PBS and not H2O2 at the same concentration as in the applied LPO system.
Author’s reply: The control sample in each experiment was PBS. This control sample did not contain any of the components of the LPO system ((pseudo)halide, hydrogen peroxide, LPO enzyme). This approach made it possible to use statistical tests to detect differences in the parameters tested between the sample treated with the system and the untreated sample (PBS only). In addition, since the lactoperoxidase system consists of three components in each experiment, we checked whether each of these components alone could influence the parameter under investigation in any way. Thus, in all experiments, samples containing hydrogen peroxide alone were compared with the control, which was always PBS. All results for all parameters tested can be found in the S1-S5 tables in the manuscript-supplementary.docx file. We hope that this additional explanation has removed any doubts about the controls that were used.
